

# Exosomal cargoes in OSCC: current findings and potential functions

Chengzhi Zhao[1,*], Geru Zhang[1,*], Jialing Liu[2], Chenghao Zhang[2], Yang Yao[3] and Wen Liao[4]

[1] State Key Laboratory of Oral Diseases & National Clinical Research Center for Oral Diseases, West China School of Stomatology, Sichuan University, Chengdu, China
[2] State Key Laboratory of Oral Diseases & National Clinical Research Center for Oral Diseases, Department of Orthodontics, West China School of Stomatology, Sichuan University, Chengdu, Sichuan, China
[3] State Key Laboratory of Oral Diseases & National Clinical Research Center for Oral Diseases, Department of Oral Implantology, West China Hospital of Stomatology, Sichuan University, Chengdu, Sichuan, China
[4] State Key Laboratory of Oral Diseases & National Clinical Research Center for Oral Diseases, Department of Orthodontics, West China Hospital of Stomatology, Sichuan University, Chengdu, Sichuan, China
[*] These authors contributed equally to this work.

## ABSTRACT

Oral squamous cell carcinoma (OSCC) is the most prevalent malignancy in head and neck cancer, with high recurrence and mortality. Early diagnosis and efficient therapeutic strategies are vital for the treatment of OSCC patients. Exosomes can be isolated from a broad range of different cell types, implicating them as important factors in the regulation of human physiological and pathological processes. Due to their abundant cargo including proteins, lipids, and nucleic acids, exosomes have played a valuable diagnostic and therapeutic role across multiple diseases, including cancer. In this review, we summarize recent findings concerning the content within and participation of exosomes relating to OSCC and their roles in tumorigenesis, proliferation, migration, invasion, metastasis, and chemoresistance. We conclude this review by looking ahead to their potential utility in providing new methods for treating OSCC to inspire further research in this field.

## INTRODUCTION

Extracellular vesicles (EVs) are membrane-bound organelles actively secreted by cells, which can be released by nearly all cell types and detected in many body fluids (*Shah, Patel & Freedman, 2018*; *Shao et al., 2018*). The various types of cargo, including proteins, lipids, and nucleic acids within and on EVs surface (*Mashouri et al., 2019*), vary depending on their parental cells of origin and extracellular environment. This is why they are regarded as a mechanism for intercellular communication (*Shah, Patel & Freedman, 2018*; *Van Niel, D'Angelo & Raposo, 2018*). Based on their morphological features and contents, EVs are classified into different groups namely microvesicles (MVs), exosomes, and apoptotic bodies (*Gyorgy et al., 2011*). Different EV subtypes are thought to be divided based on their physical characteristics (such as size or density) relying on specific secretion mechanism (*Mathieu et al., 2019*; *Théry et al., 2018*). Exosomes are one kind of EV with

Corresponding author
Wen Liao, liaowenssw@126.com

a diameter of <150 nm (*Shah, Patel & Freedman, 2018*). They origin from intraluminal vesicles (ILVs) secreted by multivesicular bodies (MVBs) through fusing with the plasma membrane (*Van Niel, D'Angelo & Raposo, 2018*). Exosomes have been shown to play significant roles in regulating physiological and pathological processes, in addition to having great potential in therapeutic development (*Ferguson & Nguyen, 2016*; *Kalluri & LeBleu, 2016*; *Liao et al., 2019*; *Mashouri et al., 2019*). Exosomes play important function in many diseases (*Hadavand & Hasni, 2019*; *Kadota et al., 2016*; *Li, Liu & Cheng, 2019*; *Malm, Loppi & Kanninen, 2016*; *Zhan et al., 2019*). In cancer, their participation in numerous process phases has been observed, including in situ tumorigenesis (*Milman, Ginini & Gil, 2019*), tumor growth (*Matei, Kim & Lyden, 2017*; *Xue et al., 2017*), angiogenesis (*Zeng et al., 2018*), evasion of immune system (*Chen et al., 2018*; *Kulkarni et al., 2019*), resistance to chemotherapeutic agents (*Mashouri et al., 2019*), and metastasis (*Hoshino et al., 2015*; *Kulkarni et al., 2019*; *Mashouri et al., 2019*). In addition, antitumor effects have also been observed in exosomes (*Pakravan et al., 2017*; *Xie et al., 2019*).

OSCC, usually preceded by white or red mucosal changes known as leukoplakia or erythroplakia, respectively, or sometimes a combination of red and white features, is the most prevalent malignancy of the head and neck (*Chi, Day & Neville, 2015*) characterized by high recurrence and poor prognosis. There are approximately 350,000–400,000 new cases each year with high risk of recurrence (20% to 30%) (*Gupta, Johnson & Kumar, 2016*). Recently, a close association between OSCC and exosomes biology has been reported. Exosomes can transport between cells and microenvironment and promote the initiation and progression of OSCC directly (*Momen-Heravi & Bala, 2018*; *Razzo et al., 2019*); they also modulate OSCC by regulating the immune system, causing metabolic dysfunction and chemoresistance (*Cui et al., 2020*). They have been developed for applications in the clinic, including as biomarkers for early diagnosis and drug delivery (*De Jong et al., 2019*; *Surman et al., 2019*; *Zlotogorski-Hurvitz et al., 2016*). In this manuscript, we reviewed all these exciting advances. By looking ahead to exosomes' potential utility in providing new methods for treating OSCC, we intend to inspire further research in this field.

## SURVEY METHODOLOGY

We systematically searched with PubMed Advanced Search Builder with the following keywords: (1) OSCC and exosomes, (2) HNSCC and exosomes, (3) proteins in exosomes and OSCC, (4) lipid and exosomes and OSCC, (5) nucleic acids and exosomes in OSCC, (6) non-coding RNA and exosomes in OSCC, (7) mitochondrial DNA and exosomes in OSCC, (8) exosomes and clinic use and OSCC. By reading the titles and abstracts, papers nonrelated with either exosomal cargoes or OSCC or HNSCC were excluded. Besides, papers not published in English were excluded. Our search was not refined by publishing date, journal or impact factor of the journal, authors or authors affiliations.

### Exosomes biogenesis

The biogenesis and release of exosomes to the extracellular environment is an ordered process. The first step for exosomes biogenesis is the formation of early endosomes (EEs). By inward budding or endocytosis, primary endocytic vesicles and their contents fuse with

each other to form EEs (*Huotari & Helenius, 2011*; *Liao et al., 2019*). After the process of primary endocytic vesicles delivering their contents and membranes to EEs in the peripheral cytoplasm over 8–15 min, EEs will then go to their destination (*Huotari & Helenius, 2011*). One possible destination for EEs is recycling to the plasma membrane, directly or with the help of recycling endosomes (*Huotari & Helenius, 2011*; *Morelli et al., 2004*). The other possibility is their conversion to late endosomes (LEs) for additional processing. LEs accumulate ILVs formed by the inward budding of the endosomal membrane (*Abels & Breakefield, 2016*), which results in the conversion of LEs to MVBs (*Abels & Breakefield, 2016*; *Liao et al., 2019*). ILV formation is exquisitely regulated by mechanisms that remain to be fully elucidated, but it is reported to mainly rely on endosomal-sorting complex required for transport (ESCRT)-dependent machinery. Multiprotein ESCRT complexes are composed of ESCRT-0, ESCRT-I, ESCRT-II, and ESCRT-III. These localize on the cytoplasmic side of the endosomal membrane and play unique roles in sorting proteins into ILVs with assistance of accessory proteins (ALIX, VPS4, and VTA1) (*Trajkovic et al., 2008*). Besides ESCRT-dependent pathway, some evidence revealed that there are alternative, independent pathways regulating ILV biogenesis (*Mashouri et al., 2019*; *McNally & Brett, 2018*; *Trajkovic et al., 2008*). The type of cargo that is sorted into ILVs may relate to the mechanisms involved. With MVBs, there are two notable fates: to either degrade by fusing with lysosomes, or having their ILVs released as exosomes by fusion with the plasma membrane (*Grant & Donaldson, 2009*; *Hessvik & Llorente, 2018*).

Exosomes biogenesis starting from the formation of EEs to the final release into extracellular environment is a complex process (Fig. 1). To our knowledge, it can differ between cell types and their cellular physiological or pathological status (*Kalluri & LeBleu, 2016*; *Liao et al., 2019*), but much remains to be discovered to fully exploit the potential for exosomes as future clinical therapies (*Mashouri et al., 2019*).

## Exosomes capsuled protein in OSCC

It is known that exosomes encapsulate genetic material as well as proteins from their cells of origin. Traditionally, the proteins that are encapsulated or transported by exosomes include structural and functional proteins that maintain the basic structure of exosomes and are responsible for regulating fusion, migration, and adhesion to target cells, such as transmembrane protein families (CD9, CD63, CD81 and CD82), molecular chaperones (Hsp70, Hsp90), multi-capsule synthesis proteins (TSG101 and ALIX), membrane carried fusion proteins, and others (*Hemler, 2003*; *Li et al., 2017a*; *Li et al., 2017b*; *van Niel et al., 2006*). Some exosomes in cancer tissues can carry specific proteins that allow their identification and differentiation from other exosomes, which could help to determine their unique function and potentially act as biomarkers for detecting disease progression (*DeRita et al., 2017*; *Melo et al., 2015*; *Zhou et al., 2006*).

In OSCC, exosomes-loaded protein both promote the tumorigenesis of OSCC and regulate the stromal cells around cancer tissues and support the processes of cancer cell proliferation (Fig. 2).

Numerous studies have already revealed that epidermal growth factor receptor (EGFR) plays an important role in tumor growth progress including tumorigenesis, tumor invasion
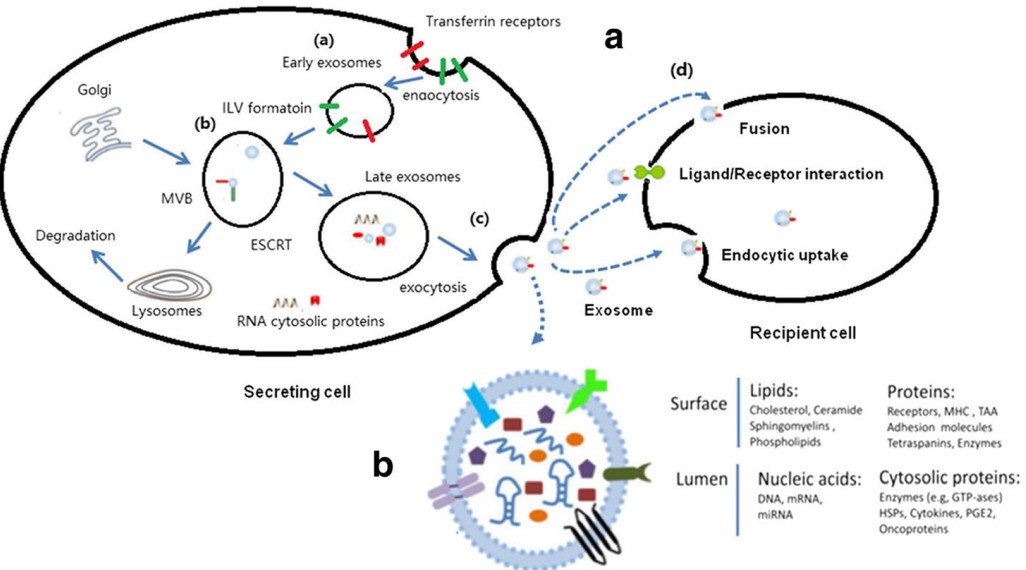

**Figure 1 Exosomes biogenesis and secretion within endosomal system.** This figure is reprinted from the manuscript *Role of exosomal proteins in cancer diagnosis* by *Li et al. (2017b)* according to open access licence CC BY 4.0.

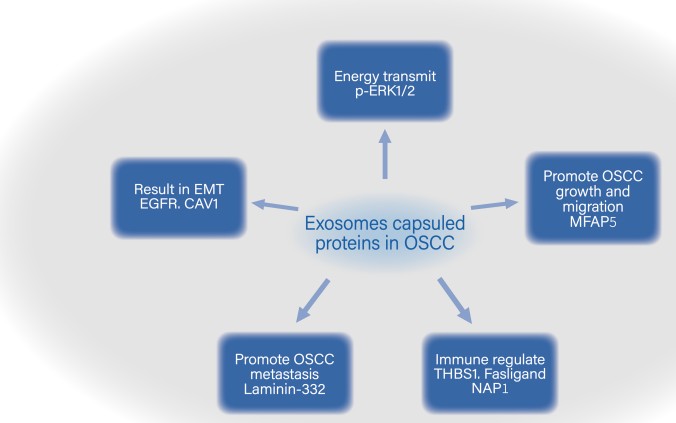

**Figure 2 Functions of exosomes capsuled proteins in OSCC.**

and drug resistance (*Shostak & Chariot, 2015*; *Sigismund, Avanzato & Lanzetti, 2018*; *Yang & Chang, 2018*). In OSCC, there is no exception. Overexpression of EGFR activates many signal pathways, such as RAS/MEK/ERK, PI3K/AKT, and JAK/STAT pathways (*Byeon, Ku & Yang, 2019*; *Nakamura et al., 2019*; *Oliveira-Silva et al., 2016*; *Zhang et al., 2018*). Secretion of EGFR-contained exosomes from OSCC can be increased by EGF's stimulation. Vesicles internalized by healthy epithelial cells around cancer cells result

in epithelial-mesenchymal transition (EMT), turning normal cell into spindle-like cells, which promotes invasion and migration of the cells within the milieu of tumor (*Aiello et al., 2018*). Fujiwara et al. established a novel mode of tumor therapy with anti-EGFR antibody cetuximab. This drug inhibits EMT in oral epithelial cell by preventing the internalization of OSCC secreted exosomes into epithelial cells (*Fujiwara et al., 2018*). In the milieu of tumor, fibroblast is another crucial portion for its participation in oral cancer, and evidences show that exosomes can help it function better. Principe et al. compared fibroblasts from resected tumor, adjacent normal tissue in the same patient and conducted a detailed analysis of the cancer associated fibroblasts (CAF) secretome in OSCC. They found that CAF-derived exosomes loaded with cytokines, growth factors or chemokine were richer in tumor tissue than normal tissues, and exosomes play important role in promoting OSCC growth and migration. The major cargo in OSCC exosomes is microfibrillar-associated protein 5 (MFAP5), which is suggested to be a novel prognostic of OSCC and exosomes loaded with those content can be a new treat target (*Principe et al., 2018*). Jiang et al. found that OSCC cells can secrete exosomes with p-ERK1/2 to adjacent normal fibroblasts, directly active the ERK1/2 signal pathway to down-regulate the expression of Caveolin-1 (CAV1) protein and up regulate the expression of MCT4 and PDK1 in normal fibroblasts. This regulation takes place both in vivo and in vitro and because of it, fibroblasts output much more lactate and cancer cells intake their outputted lactate, which then activates MCT4/MCT1 axis between cancer cell and activated fibroblasts, and, finally, provides sufficient energy for tumor cell growth and reproduction (*Jiang et al., 2019*).

Interaction between tumor cells and immune system can play a dual role: on one hand, immune cells can recognize and eliminate tumor cells in the early stages of tumor development (*Borst et al., 2018*; *Gardner & Ruffell, 2016*; *Teng et al., 2015*); on the other hand, excessive infiltration of immune cells is related with poor prognosis (*Gonzalez, Robles & Werb, 2018*; *Mantovani et al., 2017*). In the microenvironment of tumor, exosomes can mediate the reaction of immune cells (Fig. 2). Kim et al. compared exosomes from normal and OSCC patients. They found that nearly 78% of OSCC patients' serum-derived exosomes express Fas ligand (FasL) and only 5% exosomes in normal people express FasL. Moreover, co-culturing FasL(+) OSCC serum-derived exosomes with Jurkat cells lead to the apoptosis of immune cells (*Kim et al., 2005*). Zhang's group discovered that exosomes derived from supernatant of human oral cancer cells CAL 27 and SCC-25 (ATCC) were rich in interferon regulatory factor 3 (IRF-3) and its phosphorylation which promotes chemokine (C-X-C motif) ligand (CXCL) genes and the expression of the type I interferon (IFN) gene. This strengthenes the biological function of NK cells, including cytotoxicity, proliferation and release of granzyme M and perforin (*Wang et al., 2018b*). Chen and colleagues found that thrombospondin 1 (THBS1) was transported by OSCC-derived exosomes and could be taken into surrounding macrophages derived from THP-1 and PBMCs resulting in stimulating macrophage transformation into M1-like tumor associated macrophages (TAMs), this process plays a crucial role in controlling OSCC cell migration (*Xiao et al., 2018*). Besides, recent research indicates that transforming growth factor-beta (TGF-$\beta$) is important for tumorigenesis and immunosuppression in the tumor microenvironment (*Batlle & Massagué, 2019*); however, in OSCC, mutation of loss function

in TGF$\beta$ type II receptor (T $\beta$RII) is quite common. South and colleagues discovered that exosomes extract from stromal fibroblasts isolated from OSCC patients contains T $\beta$RII, they can increases TGF$\beta$ signaling in OSCC cells devoid of T$\beta$RII, which active TGF-$\beta$ signaling between tumor cells and their surrounding microenvironment (*Languino et al., 2016*). They also showed that OSCC-derived exosomes were loaded with the C-terminal fragment of desmoglein 2, a highly expressed protein in many kinds of malignant tumors. DSG2 expression in OSCC could promote exosomes secretion through the modification of metalloproteases and repartitioning of Cav1; moreover, these DSG2-CTF-positive exosomes could modulate the microenvironment by converting nearby fibroblasts into TAMs (*Overmiller et al., 2017*).

In the past few years, several treatments have been conducted to the cancer patients, metastasis is still the major cause of death in most cases. Exosomes and their carriers secreted by cancer cells would participate in pathological generation of blood and lymphatic vessels, which is closely related to the development and progression of tumor tissues (*Fu et al., 2019*; *Mashouri et al., 2019*). Vascular endothelial growth factor C (VEGF-C) is universally acknowledged as one of the most effective proteins in promoting lymphatic vessels. It is abundant in lymph node positive metastatic OSCC patients (*Lien et al., 2018*; *Shigetomi et al., 2018*) (Fig. 2). *Zhong et al. (2019)* discovered that increased VEGF-C expression associated with an increased number of salivary exosomes in OSCC tissues. In Kozaki and colleagues' study, they found that the increase of HSP90-rich exosomes is positively correlated to the increased invasive capacity of OSCC; moreover the knockdown of HSP90$\alpha$ and HSP90$\beta$ decreased the metastatic capacity and survivability of OSCC cells (*Ono et al., 2018*). Observing that OSCC LN1-1 cells were more aggressive in lymphatic node metastasis than OEC-M1 cells, *Wang et al. (2019a)* used stable isotope amino acid labeling to reveal that higher laminin-332 protein levels in tumor cell-derived exosomes was a major cause for the superior lymphangiogenesis ability of OSCC LN1-1 cells, as laminin-332 promotes lymphatic endothelial cell migration and tube formation. *Li et al. (2019a)* reported that PF4V1, CXCL7, F13A1, and ApoA1 could affect OSCC lymph node metastasis, and the mechanism remains unknown.

## Lipid cargo in OSCC exosomes

Exosomes are an alternative means to carrier proteins and lipoproteins for transporting lipids (*Record et al., 2014*). There are approximately 2000 lipid species identified through comparative lipidomic analyses (*Haraszti et al., 2016*), which can localize to the membrane and lumen of exosomes. The roles for several kinds of lipids have been reported, including BMP (Bismonoacylglycerophosphate), cholesterols, ceramides, and phosphatidic acid (*Record et al., 2014*).These lipids play critical roles in exosomes biogenesis and release (*Record et al., 2014*; *Record et al., 2018*). And some of them can be unique and help us distinguish different EVs. For example, BMP is recognized as a lipidic molecule required for MVB formation (*Subra et al., 2007*) and ILV biogenesis (*Falguières, Castle & Gruenberg, 2012*), however, it is irrelevant to the formations of MVs, which indicates its potential as a biomarker to distinguish exosomes from MVs (*Record et al., 2018*). Lipid composition is not only different between various EVs, the lipid contents of exosomes are usually

different from their parental cells. *Llorente et al. (2013)* quantified 280 species of lipids from PC-3 prostate cancer cells and their exosomes and found some differences in their lipid composition. This interesting finding indicates that the discrimination of lipids between exosomes and parental cells could play significant roles in many pathophysiologies, which inspires us that by detecting the lipid composition from exosomes and parental cells, it can be a diagnostic as well as therapeutic direction in the future. Over the past few years, lipid metabolic abnormalities have been identified as a feature of tumor cells. *Hu et al. (2019)* found that some lipid metabolism-related genes in OSCC patients could be used for prognostication. Lipid uptake can also be enhanced in cancer (*Tousignant et al., 2019*). *Pascual et al. (2017)* observed a similar phenomenon in OSCC, as they documented a subpopulation of CD44(bright) OSCC cells that express high levels of lipid metabolic genes and can initiate metastasis.

Arachidonic acid (AA) is a free fatty acid that can be transported by exosomes (*Subra et al., 2010*), and its metabolism is a major dysregulated pathway in cancer cells (*Gatto, Schulze & Nielsen, 2016*). As the precursor of both leukotrienes and prostaglandins, AA can be transferred between tissues by exosomes contributing to tumor growth and progression (*Linton et al., 2018*). Exosomes secreted from AsPC-1 cells, a highly metastatic pancreatic ductal adenocarcinoma cell line, were reported to deliver AA to macrophages. The fusogenicity of AsPC-1 exosomes decreased when pretreatment with PLA2 caused the removal of AA, indicating that exosomal AA may enhance crosstalk between cancer cells and TAMs, thus contributing to tumor progression (*Linton et al., 2018*). Besides AA itself, its products have been detected performing similar functions. Leukotrienes (LTs) are involved in various pathophysiologies such as inflammatory asthma, atherosclerosis, and cancer (*Record et al., 2014*). Several types of LTs, such as LTB4, LTC4, cysteinyl leukotriene LTC4, and LTC4 synthase, are enriched in exosomes (*Esser et al., 2010*; *Lukic et al., 2019*). Exosomes appear to play a role in the biogenesis of LTs. For example, LTA4, the precursor of leukotrienes, only has a five-second half-life in in vitro buffer, but with the protection of exosomes, their half-life can be elongated to several minutes (*Esser et al., 2010*). Besides assisting the biogenesis of LTs, exosomes seem to play other roles helping LTs performing their pathological functions. Lukic et al. found that exogenous LTC4 generated by monocytic cells can be transformed into pro-tumorigenic LTD4 through gamma-glutamyl transpeptidase-1(GGT-1). GGT-1 is contained within both exosomes and primary cancer cells, which stimulates cancer cell migration and survival (*Esser et al., 2010*). And since the 5-lipoxygenase (5-Lox) and cyclooxygenase (COX)-2 pathways of arachidonic acid metabolism are involved in oral carcinogenesis (*Lai et al., 2018*), LTs in exosomes may also play an important role in OSCC. Prostaglandins (PGs) are another product of AA. Exosomes with high concentrations of prostaglandins transport more PGE2 to neighboring cells, enhancing the overall presence of PGE2 in the microenvironment (*Subra et al., 2010*). Cell motility and metastatic status are impacted by extracellular levels of PGE2 (*Nasry, Rodriguez-Lecompte & Martin, 2018*). Evidence has shown that exosomes rich in PGE2 participate in tumor immune evasion and promote tumor growth (*Record et al., 2014*). PGE2-mediated inflammation contributing to OSCC at different stages of carcinogenesis, invasion and metastasis contributes to patient morbidity and

mortality (*Harada et al., 2017*). Abrahao et al. showed that HNSCC cells secrete PGE2. The overexpression of COX-2 in tumor and inflammatory cells, and subsequent increased production of PGE2, may promote HNSCC growth in an autocrine and paracrine way in the microenvironment. Exosomes can therefore be considered a potential medium for autocrine and paracrine regulation (*Luga et al., 2012*).

## Nucleic acids in OSCC exosomes

A diverse collection of nucleic acids, including DNA, coding mRNA, non-coding RNA (ncRNA), micro RNA (miRNA), circular RNA (circRNA), and long non-coding RNA (lncRNA) have been identified in exosomes (*Abels & Breakefield, 2016*). The nucleic acid content of exosomes is believed to play a significant role in promoting cancer pathogenesis through the oncogenic transformation and transfer of cancer-specific genetic material (*Kalluri & LeBleu, 2016*). Understanding how nucleic acids transported by exosomes mediate the process of OSCC is critical, and could illuminate strategies for exosomes-based targeted therapy.

### *DNA*

DNA in exosomes has been observed in cell culture supernatant as well as human and mouse biological fluids such as blood, seminal fluid, and urine (*Kalluri & LeBleu, 2016*). Based on cell origin, it is likely that different types of exosomes contain distinct types of DNA, such as single-stranded DNA (ssDNA), double-stranded DNA (dsDNA), mitochondrial DNA (mtDNA), and of varying states (e.g., fragment length, chromosome-bound) (*Kalluri & LeBleu, 2016*). Among these various types, dsDNA is the most evaluated (*Kahlert et al., 2014*).

Exosomal DNA has been found to be involved in immunity regulation (*Diamond et al., 2018*; *Kitai et al., 2017*). When tumors were treated by different strategies (e.g., antitumor drugs or radiotherapy), the induced secretion of cancer cell-derived exosomes containing DNA triggered dendritic cell (DC) activation and cytokine production, both of which can have antitumor effects by regulating immune responses (*Diamond et al., 2018*; *Kitai et al., 2017*). In addition to its potential therapeutic roles by regulating immune system, DNA in exosomes may become an attractive candidate biomarker for tumor diagnosis because of the inherent stability of DNA within exosomes (*Kalluri & LeBleu, 2016*). For example, the same mutations in susceptibility genes were found in exosomal DNA and parental cells of pheochromocytomas and paragangliomas (*Wang et al., 2018a*), and mutant KRAS, TP53, NOTCH1, and BRCA2 DNA in exosomes from pancreatic cancer were also detected (*San Lucas et al., 2016*; *Yang et al., 2017a*). Exosomes in OSCC may play similar roles, as mutant genes were discovered in OSCC cells as well (*Biswas et al., 2019*; *Natsuizaka et al., 2017*).

Virus infection is closely related to cancers including OSCC. Moreover, viral DNA has also been detected in exosomes from cancer patients (*Yang et al., 2017b*). For example, *Meckes Jr et al. (2010)* identified that ERK and PI3K/AKT signaling pathways can be activated if the recipient cells were exposed to exosomes containing major EBV oncogene LMP1. Besides, Human papillomavirus (HPV) is considered a risk factor for OSCC, and its DNA has been found in exosomes from HeLa cells (*Mata-Rocha et al., 2020*). There

are some evidence demonstrating that HPV DNA in exosomes participated in the process of cancer. HPV DNA in plasma-derived exosomes was detected in rectal squamous cell carcinoma patients (*Ambrosio et al., 2019*). Ambrosio et al. isolated exosomes from the HPV DNA-positive cell line CaSki, which can transfer DNA to normal cell lines. Moreover, circulating exosomes-encapsulated HPV DNA in the blood of neoplastic patients was verified to be transferred to normal and tumor cells at least in vitro. Exosomes containing HPV are involved in HPV-associated carcinogenesis, indicating a potential role in OSCC although there is no direct evidence noted (*Guenat et al., 2017*).

Mitochondria are major energy generators in cells, and increasing studies have revealed their role in tumor development. Currently, scientists are deepening our understanding of the connections between mitochondria and tumors. For example, cancer cells can obtain mitochondria from healthy cells to achieve chemical resistance (*Pasquier et al., 2013*), and mitochondrial DAMP provides tumor cells with a possible immune escape mechanism (*Zhang, Itagaki & Hauser, 2010a*; *Zhang et al., 2010b*). The transfer of mitochondria or mtDNA is not completed independently, but is carried out by the transporting mediators between cells, such as exosomes and secretory vesicles. In oral cancer, whether exosomal mtDNA participates in tumorigenesis, tumor proliferation, and migration is still vague. However, Uzawa et al. proposed a novel method for detecting mtDNA in OSCC patients and reported significant differences in serum mtDNA levels before and after OSCC patient treatment. They also identified mtDNA as a promising molecular marker for OSCC prognostication (*Uzawa et al., 2015*). Despite these advances, further exploration into exosomes-derived mtDNA and its application in the diagnosis and treatment of OSCC is necessary.

### mRNA

Coding mRNA has been found in exosomes. Transferring mRNA within exosomes can enable mRNA translation into proteins in recipient cells (*Huang et al., 2013*; *Valadi et al., 2007*), giving mRNA a potentially powerful role in cell-to-cell communication. Its function has been studied in both healthy and pathological states (*Feng et al., 2018*; *Valadi et al., 2007*). As a means for cell-to-cell transportation, the mRNA profiles of tumor-derived exosomes have been evaluated in various cancers such as melanoma (*Del Re et al., 2018*), glioblastoma (*Skog et al., 2008*), prostate cancer (*Lázaro-Ibáñez et al., 2017*), and colorectal cancer (*Hao et al., 2017*). Moreover, these studies revealed that mRNAs in tumor-derived exosomes can play vital roles in promoting malignant tumor growth, proliferation, and metastasis through suppressing immune responses and resisting antitumor treatment (*Del Re et al., 2018*; *Geis-Asteggiante et al., 2018*; *Skog et al., 2008*).

Salivary liquid biopsy has emerged as an excellent method for disease detection, which is also applied to OSCC (*Yakob et al., 2014*). Evaluation of the salivary transcriptome from OSCC patients has helped to identify some mRNA biomarkers (*Brinkmann et al., 2011*). Qadir et al. characterized and compared the transcriptome profiles between exosomes isolated from primary human normal oral keratinocytes (HNOK) and HNSCC cell lines. The results showed that in HNSCC-derived exosomes, the expression of matrix remodeling (EFEMP1, DDK3, SPARC), cell cycle (EEF2K), membrane remodeling (LAMP2, SRPX),

differentiation (SPRR2E), apoptosis (CTSC), and transcription/translation (KLF6, PUS7) factors showed significant differences from healthy cell-derived exosomes (*Qadir et al., 2018*), indicating that cancer cells may confer transcriptome reprogramming through exosomes to enhance cancer-associated pathologies.

### ncRNA

Most of the human genome is considered biologically active. However, only a minor fraction of DNA encodes proteins. ncRNAs represent the majority of RNA that is not translated into proteins (*Romano et al., 2017*). NcRNAs are a category of exosomal cargo under investigation for its complex role in regulating gene expression (*Zhang et al., 2015*). NcRNA interactions are often interconnected which, when deregulated, could eventually drive tumorigenesis and progression (*Chan & Tay, 2018*). Identifying ncRNAs and their interactions will help to provide robust biomarkers and new therapeutic targets for more effective cancer therapies, better outcomes, and greater survival (*Chan & Tay, 2018*; *Zhang et al., 2015*) (Table 1).

### miRNA

MiRNA are small ncRNAs around 22 nucleotides long, and they can be divided into oncogenic miRNA and tumor suppressor miRNA (*Svoronos, Engelman & Slack, 2016*). They perform their post-transcriptional regulatory effects by binding to specific sites known as miRNA response elements (MREs) on their target transcripts, leading to either transcript degradation or translational inhibition (*Chan & Tay, 2018*). MiRNA regulatory activity in cancer has been widely studied (*He et al., 2020*; *Li et al., 2019b*; *Sakha et al., 2016*).

*Exosomal miRNAs in OSCC growth.* MiRNA function in OSCC has been thoroughly discussed (*Langevin et al., 2017*). Since they are secreted from various types of healthy and tumor cells (*Tkach & Théry, 2016*), miRNAs in exosomes can play different roles in promoting or inhibiting cancer. Dickman et al. found that miR-142-3p secreted from oral cancer cells promotes cancer cell growth by eliminating the miRNA tumor suppressive effect. Exosomes also promote tumor angiogenesis by releasing miR-142-3p to its microenvironment (*Dickman et al., 2017*). However, in another study of *Higaki et al. (2020)*, overexpression of miR-6887-5p in SCC/OSCC cells inhibited tumor growth. Similar results document targeting miRNA as a treatment strategy to inhibit tumor growth. Lower levels of miR-3188 were detected in CAFs than normal fibroblasts, and loss of miR-3188 promoted malignant phenotypes in head and neck cancer cells, supporting its consideration as a therapeutic target (*Wang et al., 2019b*).

*Exosomal miRNAs in OSCC cell migration and invasion.* MiRNA in exosomes not only promote or inhibit tumor growth, but have also been shown to participate in OSCC cell migration and invasion. A research compared miRNA profiles in non-invasive SQUU-A and highly invasive SQUU-B tongue cancer cell clones, it was observed that hsa-miR-200c-3p acts within a key pro-invasion role in OSCC. The transfer of miR-200c-3p in exosomes derived from a highly invasive OSCC line can also accelerate the invasion potential of non-invasive counterparts (*Kawakubo-Yasukochi et al., 2018*). Another research done by

Zhao et al. (2020), *PeerJ*, DOI 10.7717/peerj.10062

**Table 1** NcRNAs regulating the process of OSCC in exosomes.

| NcRNAs | Types of ncRNAs | Pro/ Anti-tumor | Target/ Signal pathway | Functions | Origin of exosomes | Ref. |
|---|---|---|---|---|---|---|
| miR8485 | miRNA | Pro-tumor | — | Promote the carcino-genesis of premalig-nant lesions, proliferation, migration and invasion of tumor cells | MSCs | *Li et al. (2019b)* |
| miR-6887-5p | miRNA | Anti-tumor | HBp17/FGFBP-1 | Inhibit tumor cell proliferation, colony formation, then tumor growth | A431 cells | *Higaki et al. (2020)* |
| miR-142-3p | miRNA | Pro-tumor | TGFBR1 | Cause tumor-promoting changes | Oral dysplasia and OSCC cell lines | *Dickman et al. (2017)* |
| miR-24-3p | miRNA | Pro-tumor | PER1 | Maintain the proliferation of OSCC cells | Saliva in OSCC patients | *He et al. (2020)* |
| miR-3188 | miRNA | Anti-tumor | BCL2 | The loss of miR-3188 in exosomes contributes to the malignant phenotypes of HNC cells through the depression of BCL2 | CAFs | *Wang et al. (2019a)* *Wang et al. (2019a)* |
| miR-34a-5p | miRNA | Anti-tumor | AXLAKT/GSK-3beta/ beta-catenin signaling pathway | MiR-34a-5p binds to direct downstream target AXL to suppress OSCC cell proliferation and metastasis | CAFs | *Li et al. (2018a)*, *Li et al. (2018b)*, *Li et al. (2018c)* |
| miR3825p | miRNA | Pro-tumor | — | Responsible for OSCC cell migration and invasion | CAFs | *Sun et al. (2019)* |
| miR-21-5p | miRNA | Pro-tumor | — | Increase metastasis, stemness, chemoresistance and poor survival in patients with OSCC | CAL27 and SCC-15 OSCC cells | *Chen et al. (2019)* |

Zhao et al. (2020), *PeerJ*, DOI 10.7717/peerj.10062

**Table 1** (*continued*)

| NcRNAs | Types of ncRNAs | Pro/ Anti-tumor | Target/ Signal pathway | Functions | Origin of exosomes | Ref. |
|--------|-----------------|-----------------|------------------------|-----------|---------------------|------|
| miR-1246 | miRNA | Pro-tumor | DENND2DERK/ AKT pathway | Increase cell motility and invasive ability | HOC313-LM OSCC cells | *Sakha et al. (2016)* |
| | | Pro-tumor | miR-21/HIF-1alpha/ HIF-2alpha-dependent pathway | MiR-21 can be delivered to normoxic cells to promote prometastatic behaviors | Hypoxic OSCC cells | *Li et al. (2016)* |
| miR-21 | miRNA | | | | | |
| | | Pro-tumor | PTEN, PDCD4 | Induce cisplatin resistance of OSCC cells | HSC-3-R and SCC-9-R | *Liu et al. (2017)* |
| miR-200c-3p | miRNA | Pro-tumor | CHD9, WRN | Spread invasive capacity by exosomes in tumor microenvironment | SQUU-B tongue cancer cell clones | *Kawakubo-Yasukochi et al. (2018)* |
| miR-155 | miRNA | Pro-tumor | — | Lead to mesenchymal transition and increase migratory potential and acquire cells drug-resistant phenotype | Cisplatin resistant OSCC cells | *Kirave et al. (2020)* |
| miR-200c | miRNA | Anti-tumor | TUBB3, PPP2R1B | Increase the sensitivity of Docetaxel (DTX) resistant HSC-3 cells to DTX | normal tongue epithelial cells (NTECs) | *Cui et al. (2020)* |
| miR-101-3p | miRNA | Anti-tumor | COL10A1 | Overexpression of miR-101-3p inhibit oral cancer progression and provide a therapeutic target | human bone marrow mesenchymal stem cells (hBMSCs) | *Xie et al. (2019)* |
| miR-29a-3p | miRNA | Pro-tumor | SOCS1 | Promote M2 subtype macrophage polarization, tumor cell proliferation and invasion | SCC-9 and CAL-27 | *Cai et al. (2019)* |
| FLJ22447 | lncRNA | Pro-tumor | Lnc-CAF/IL-33 | Reprogram normal fibroblast to CAFs and promote OSCC development | CAFs | *Ding et al. (2018)* |

Sun et al. also found that miR3825p was overexpressed in CAFs compared with fibroblasts of adjacent normal tissue, and miR3825p overexpression was an important regulatory factor in OSCC cell migration and invasion (*Sun et al., 2019*). MiRNAs in exosomes can also play like messengers to modulate tumor environment, and then manipulate OSCC cell migration and invasion. Normoxic and hypoxic OSCC-derived exosomes yielded different miRNA profiles, miR-21 showed its most significant role under hypoxic conditions. The loss of miR-21 in hypoxic OSCC cells downregulated miR-21 levels in exosomes and significantly reduced cell migration and invasion. Restoration of miR-21 expression in HIF-1 $\alpha$- and HIF-2 $\alpha$-depleted exosomes rescued OSCC cell migration and invasion (*Li et al., 2016*).

*Exosomal miRNAs in OSCC metastasis.* Cancer metastasis is a very complicated process, involving intrinsic and extrinsic mechanisms. Metastasis is also a great challenge in our effort to fight cancer, and it caused poor prognosis of OSCC (*Sun et al., 2019*). It has been explosively studied and proved miRNAs participated in this process of OSCC. Some studies have noticed that cancer stem cell-derived extracellular vesicles are enriched with miR-21-5p, which is associated with increased potential of OSCC metastasis (*Chen et al., 2019*). However, some miRNAs have been revealed as performing anti-cancer roles as well. By examining the miRNA profiles of CAF- and normal fibroblast (NF)-derived exosomes, miR-34a-5p expression was found to be significantly decreased, making it an anti-cancer therapeutic target for OSCC (*Li et al., 2018c*). The complex of metastasis makes it a fascinating field, especially the relations between miRNAs and tumor microenvironment (TME). TME plays a vital role in the progression of OSCC. Recent research has revealed that tumor-derived exosomes (TEX) accumulate in the TME and interact between tumor and healthy stromal cells (*Ludwig et al., 2018*).Cai et al. cocultured exosomes extracted from OSCC cell lines (SCC-9 and CAL-27) with macrophages (*Cai et al., 2019*). Their results showed that the upregulation of miR-29a-3p in OSCC-derived exosomes is related to M2 subtype macrophage polarization. After interfering with miR-29a-3p from OSCC, M2 subtype macrophage polarization was inhibited by OSCC-derived exosomes.

*Exosomal miRNAs in chemoresistance.* Chemotherapy is a hallmark of fighting cancers. However, Chemoresistance is a significant challenge for OSCC treatment with no clear mechanism. Several studies have shown that miRNAs in exosomes of both healthy and tumor cells can manipulate this phenomenon (*Cui et al., 2020*; *Kirave et al., 2020*; *Qin et al., 2019*). Some miRNAs are upregulated during chemotherapy, which can enhance chemoresistance against antitumor drugs such as cisplatin (CIS) and docetaxel (DTX). *Kirave et al. (2020)* found that when transferring exosomes from CIS-resistant to CIS-sensitive cells, miR-155 was significantly upregulated in the recipient CIS-sensitive cells. Exosomes isolated from CIS-resistant cell lines contained a higher concentration of miR-21 in accordance with the parental cells' increased cisplatin resistance, which indicates that miR-21 may be a potential target against chemoresistance (*Liu et al., 2017*). The underlying mechanism is considered related to EMT and decreased DNA damage in cancer cells (*Cui et al., 2020*; *Kirave et al., 2020*). Some miRNAs were downregulated during chemoresistance.

For example, downregulation of miR-200c increased resistance to DTX, when miR-200c was transported by exosomes, the results showed the increase of the sensitivity to DTX both in vitro and in vivo, indicating miR-200c could be a therapeutic target of OSCC (*Cui et al., 2020*).

### lncRNA and circRNA

Beyond miRNA, there are other regulatory ncRNAs that perform complex roles in cancer (*Morris & Mattick, 2014*). One kind, lncRNA, ranging from 200 to >1000 nucleotides, is a novel class in the human genome with seldom to no coding potential (*Yang, Wen & Zhu, 2015*). They participate in various diseases through interacting with DNA (*Lin et al., 2020*), RNA, or proteins (*Zhang et al., 2014*). lncRNA may be involved in cancer cell proliferation (*Wang, Li & Shi, 2019*; *Xu et al., 2019*), migration (*Xu et al., 2019*), invasion (*Xiong et al., 2019*), metastasis (*Jin et al., 2020*; *Xiong et al., 2019*), and antitumor drug resistance (*Wang, Li & Shi, 2019*). LncRNA has multiple roles in regulating cancers, and the most important one among them is acting as "miRNA sponge" (*Tan, Xiang & Xu, 2019*; *Zhang et al., 2019b*). *Zhang et al. (2019b)* discovered that knockdown of lncRNA UCA1 significantly suppressed TGFβ1-induced tongue cancer cell invasion and eventually induced EMT. TGFβ1-induced EMT and invasion in OSCC are consistent with increased JAG1, whereas miR-124 inhibits its expression. UCA1 binds to miR-124 directly and can downregulate miR-124 expression. This is the basis for lncRNA UCA1's protumor effect through sponge-like lncRNA-miRNA-mRNA regulation (*Zhang et al., 2019b*). lncRNA TIRY was also found to act as a miRNA sponge in OSCC by downregulating miR-14 expression in CAF-derived exosomes (*Jin et al., 2020*). lncRNA FLJ22447 (lnc-CAF) secreted from CAFs regulates NFs to CAFs, and tumor cells increased lnc-CAF levels in stromal fibroblasts via exosomal lnc-CAF as well (*Ding et al., 2018*). Unlike lncRNA, circRNA consists of a closed continuous loop structure without 5′–3′ polarity or a poly-A tail, which enables its resistance to RNases and higher stability compared with linear RNA (*Bai et al., 2019*). Similar to lncRNA, circRNA also functions as a miRNA sponge (*He et al., 2019*). Although the roles of circRNA in exosomes remains unknown, a hypothesis has been introduced by *Bai et al. (2019)*. Some circRNAs may bind to and transport with miRNAs by exosomes. After entering target cells, miRNAs are released to regulate target genes (*Jin et al., 2020*). CircRNAs in exosomes may therefore enter the recipient cells, bind to miRNAs, and regulate target genes. The roles of circRNAs in OSCC have been investigated by several researchers (*Han, Cheng & Li, 2020*), and differences in circRNAs profiles between OSCC patients and healthy people have also been distinguished (*Qiu et al., 2019*; *Wei et al., 2020*). By performing the function of miRNA sponge, the significance of LncRNA and circRNA has been noted, their discovery in OSCC also indicates that their roles in the pathogenesis cannot be neglected. Although there is still a lack of direct evidence for exo-lncRNA and exo-circRNA regulating OSCC, the evidence presented in other cancers proved it must be a remarkable field in the future research of OSCC as well (*Li et al., 2018a*; *Li et al., 2018b*; *Ren et al., 2018*; *Zhang et al., 2019a*).

### Other ncRNAs

Besides regulatory ncRNA, tRNA and rRNA make up another group of ncRNA referred to as housekeeping ncRNA (*Romano et al., 2017*). *Baglio et al. (2015)* defined the exosomes-enclosed RNA species from the full small RNAome of MSC-produced exosomes. Adipose and bone marrow MSC subtypes secrete different tRNA species that may be relevant to clinical applications; however, how tRNAs are transported through exosomes and their influence on the microenvironment in a cell type-dependent manner remains unclear (*Baglio et al., 2015*). Crescitelli et al. analyzed RNA profiles in different EVs including exosomes. According to their findings, rRNA was primarily detectable in apoptotic bodies, but smaller RNAs without prominent ribosomal RNA peaks in exosomes (*Crescitelli et al., 2013*). This indicates that exosomes are potentially not carriers of rRNA. Collectively, there is little evidence surrounding the exosomal transportation of tRNA and rRNA, let alone their potential function in modulating cancer and their microenvironment.

## Clinical use of exosomes
### Exosomes as biomarkers for diseases diagnosis

Combining exosomes' stability and accessibility in various biological fluids like urine, blood (including serum), breast milk, exosomes can be used as biomarkers for various cancers, some can even be used to determine the type and severity of the disease. For example, exosomes carrying Glypican-1 are considered a sensitive indicator of pancreatic cancer in blood samples (*Melo et al., 2015*). In gastric cancer patients, exosomes expressing CD63 can be isolated from tumor cells cluster but not stromal cells reflects a worse prognosis than the situation that CD63+ exosomes can be find in both cell types (*Miki et al., 2018*). As for OSCC, exosomes have also been considered as one of the fast-detected methods for they have been discovered in the tumor microenvironment and been discovered play an irreplaceable role in OSCC's tumorigenesis, tumor proliferation and migration, tumor invasion and metastasis (including angiogenesis and lymphangiogenesis), chemoresistance, and so on (Fig. 3). Many studies have shown that the morphology and content of exosomes isolated from OSCC patients' saliva and blood are very different from that of healthy people, suggested the possibility of using exosomes for OSCC diagnosed (*Zlotogorski-Hurvitz et al., 2016*). By using high-resolution AFM, Sharma et al. found the salivary exosomes of oral cancer patients were much larger and amorphous compared with those of healthy people, and also found that CD63 was significantly increased on the surface of cancer exosomes (*Sharma et al., 2011*). In a study by Rabinowits, tongue squamous cell carcinoma tissue and normal tissue were collected in pairs. They isolated exosomes and found different miRNA loading patterns similar to the loading patterns of blood exosomes, suggesting that circulating exosomes can be a more reliable method in evaluating tumors (*Rabinowits et al., 2017*). *Sanada et al. (2020)* examined the expression levels of secreted lysyl-oxidase-like 2 (LOXL2) in pharyngeal and tongue cancer patient serum and found that elevated serum exosomes LOXL2 levels are associated with low-grade oral cancer. All these discoveries of exosomes provide new ideas for the non-invasive disease diagnosis method.

Also, the contents of exosomes can indicate tumor invasion capacity and occurrence of distant metastasis, because exosomes always participate in the angiogenesis and

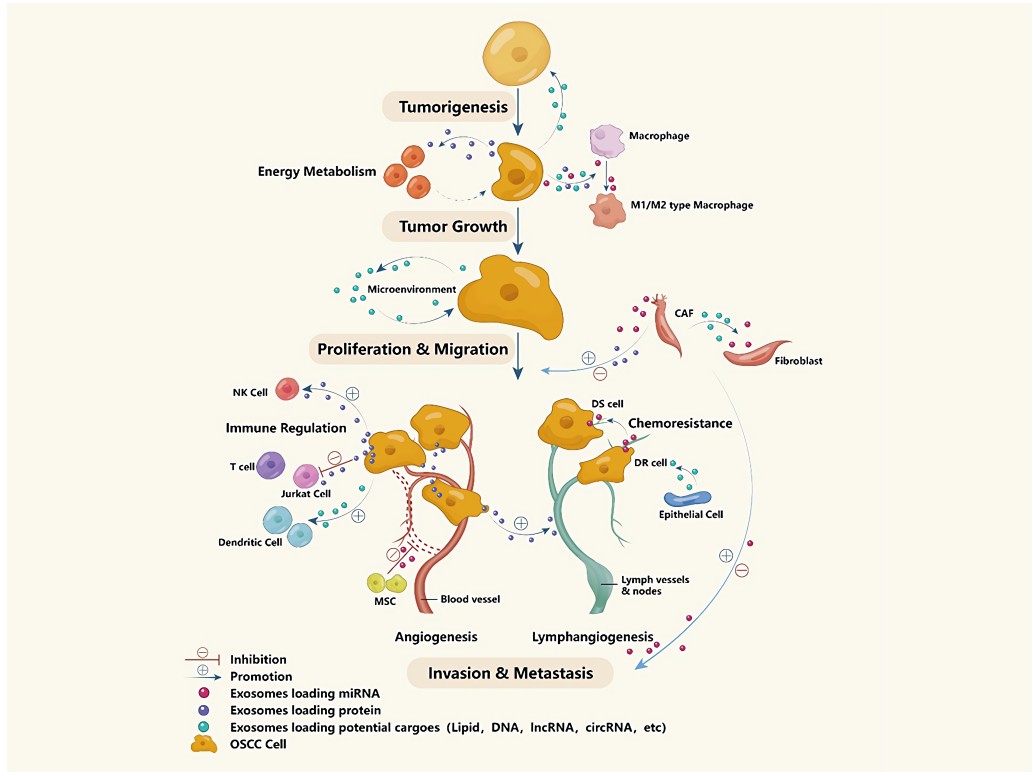

**Figure 3** Exosomes' contents and potential functions in the developing process of OSCC.

lymphangiogenesis of tumor tissues as mentioned before. Li and colleagues (*2016*) found that miR-21-rich exosomes are associated with increased OSCC invasiveness, and that these exosomes are delivered to normoxic cells to promote prometastatic behaviors. Nakashima et al. utilized integrated microarray profiling technology to analyze the different expression patterns of miRNA between non-invasive and highly invasive tongue cancer cells, observing that hsa-miR-200c-3p was the crucial point in spreading invasive ability (*Kawakubo-Yasukochi et al., 2018*). Moreover, the quantity and content of exosomes can be used in non-invasive examination for deciding tumor stages and predict the prognosis of treated OSCC patients, and scientists suggested that the combination of different kinds of biomarkers are significantly better than single biomarker in OSCC diagnosing. Ludwig et al. suggest that tumor staging can be understood by exploring the interaction between OSCC cell-derived exosomes and lymphocytes. Exosomes in the plasma of patients with tumors in an uncontrolled phase have greater induction of T cell apoptosis and inhibition of lymphocyte proliferation, which differs from patients without significant disease (*Ludwig et al., 2017*). The quantity and content of exosomes could predict the prognosis of treated OSCC patients. Liu and Tian compared serum exosomes between laryngeal squamous cell carcinoma patients and vocal cord nodule patients, finding that the expression levels of miR-21 and HOTAIR were higher in exosomes of malignant lesions. Moreover, the serum exosomes of patients with laryngeal cancer in stage III/IV also showed a high level of

miR-21 and HOTAIR in exosomes (*Wang et al., 2014*), suggesting an association between the level of exosomal content and prognosis. Those outcomes are similar to a clinical study performed by *Rodríguez Zorrilla et al. (2019)*, they collected ten OSCC patients' plasma sample before and after the excision surgery to evaluate the level of plasmatic CD63+ and CAV1+ exosomes levels correlate with patient's overall survival, they came to an conclusion that the lower expression of CD63+ exosomes after surgery and lower CAV1+ exosomes before surgery would indicate a longer life expectancy.

## Exosomes as therapeutic mediums

In terms of exosomal structure, they consist of biogenic lipid bilayers similar to cell membranes, which could protect cargo from degradation and could deliver the drug to target cells. The surface diameter of exosomes is only 40–150 nm, so they are small enough to access most tissue without consumption and degradation by macrophages (*De Jong et al., 2019*; *Kalluri & LeBleu, 2016*; *Surman et al., 2019*; *Vader et al., 2016*; *Yang & Wu, 2018*). Moreover, they may exhibit inherent targeting properties, which is determined by lipid composition and protein content (*Murphy et al., 2019*). With these advantages, they are considered as potential drug delivery systems. At present, clinical trials using exosomes as a drug delivery method against cancer have been gradually increasing. In lung cancer, tumor cell-derived exosomes were extracted from the pleural effusion of lung cancer patients. After modification and loading with the chemotherapy drug methotrexate, they were reinjected into the patient's chest cavity. It has been observed that exosomes have a safe inhibitory effect on the growth of tumor cells (*Guo et al., 2019*). In colon cancer, exosomes with carcinoembryonic antigen were isolated from ascites fluid. After combining them with granulocyte macrophage colony stimulating factor, they served as a vaccine to induce a beneficial tumor-specific antitumor cell toxic T lymphocyte response (*Dai et al., 2008*). In OSCC, the current drug-loading process is mainly based on different carrier systems, such as nanoparticles, nanolipids, and hydrogels, which can alleviate the disadvantage of poor water solubility for oral cancer anti-cancer drugs to a certain extent (*Ketabat et al., 2019*; *Luo et al., 2014*; *Poonia et al., 2017*). Studies have shown that exosomes have been used as carriers for chemotherapeutic agents such as curcumin, DOX, and PTX, thereby reducing their side effects and improving therapeutic efficiency (*Batrakova & Kim, 2015*; *Sun et al., 2010*; *Tian et al., 2014*). Despite this progress, research on exosomes as a drug-loading system is still limited, mainly due to their limited ability to deliver high-dose therapeutic drugs, insufficient basic experiments, and a lack of effective, standardized separation and purification methods (*Ludwig, Whiteside & Reichert, 2019*; *Yang & Wu, 2018*).

## CONCLUSIONS

Researchers are continuing to provide new evidences on the importance of exosomes in cancer tumorigenesis, proliferation, migration, invasion, metastasis, and chemoresistance. Exosomal cargo based drug delivery system is also interesting in the treatment of OSCC because of its numerous type and abundance of cargo found in OSCC-related exosomes, which varies among different states of health and disease. What is more, because of their unique secretion patterns, exosomes can be used as ideal biomarkers for OSCC diagnosis,

especially in saliva biopsy, which is a non-invasive method comparing to serum biopsy. With all these possibilities, however, there is still a lot not clearly understand in exosomes generation, secretion, cargo transportation, fusion with target membrane, especially the consistent test standard and separation method of exosomes has not been completely established. A more comprehensive understanding of the complexity of exosomes would help us elucidate disease mechanisms and provide opportunities for the diagnosis and treatment of OSCC. The safety of exosomes is still another question not solved yet. Further studies in this field, such as clinical study or studies using large animal models, are still needed to prove the safety of exosomes in treatment, especially for exosomes secreted from cancer cells. Despite the faced great challenges and difficulty, we still find exosomes' great potential of usage in OSCC treatment and biopsy. We believe that exosomes can play even more important roles in this field.

## ACKNOWLEDGEMENTS

We acknowledge Dr. Qiang Peng for his valuable suggestion.

### Funding

This work was supported by National Natural Science Foundation of China (No. 81700941), the Science and Technology Department of Sichuan Province (No.2020YFS0087) and Sichuan University-Luzhou City cooperation project (No.2018CDLZ-14). The funders had no role in study design, data collection and analysis, decision to publish, or preparation of the manuscript.

### Grant Disclosures

The following grant information was disclosed by the authors:
National Natural Science Foundation of China: 81700941.
Science and Technology Department of Sichuan Province: 2020YFS0087.
Sichuan University-Luzhou City cooperation project: 2018CDLZ-14.

### Competing Interests

The authors declare there are no competing interests.

### Author Contributions

- Chengzhi Zhao and Geru Zhang conceived and designed the experiments, performed the experiments, analyzed the data, prepared figures and/or tables, authored or reviewed drafts of the paper, and approved the final draft.
- Jialing Liu performed the experiments, prepared figures and/or tables, and approved the final draft.
- Chenghao Zhang analyzed the data, prepared figures and/or tables, and approved the final draft.
- Yang Yao and Wen Liao conceived and designed the experiments, authored or reviewed drafts of the paper, and approved the final draft.

## Data Availability

This is a literature review. There is no raw data.

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
