# Peer review of "Exosomal cargoes in OSCC: current findings and potential functions"

_PeerJ, doi:10.7717/peerj.10062_

## Round 0.1 · original submission · Major Revisions

The comments of the reviewers are enclosed; all found it to be an interesting area with merit and an appropriate review choice. I and two of the reviewers both feel however that the article needs a thorough reorganization. In many part of the manuscript is not easy to understand what you wish to convey to the reader; the narrative flow of the review is not strong.

The general impression is that the manuscript is just a list of information from the literature without critical discussion or review.

Could I therefore ask you to revise your article carefully with emphasis on the 'story' you wish to tell, the flow of the discussion and use carefully linking sentences to weave your review into a more cogent discussion of the field?

The journal can offer English assistance if you wish.

Reviewer 1 ·

Basic reporting

This is a nice review from Dr. Zhao et al. that summarizes the current findings and potential functions of exosomal cargoes in OSCC and their potential clinical use.

The topic is interesting and well descripted.
The manuscript is well-written and timely. Moreover, the figures are clear.

Experimental design

The manuscript flows logically.

It is very well referenced, however I would suggest the authors to add Thery 2018 (PMID 28598415) particularly relevant in the context of extracellular vesicle characterization.

I also suggest the Authors to better explain the link between CAV1, exosomes, EMT and fibroblasts (lines 188-189)

Validity of the findings

Arguments are well developed

Additional comments

A great contribute that represents a good source for extracellular vesicles community.

Reviewer 2 ·

Basic reporting

1)Exosomes in diseases especially in cancer attract great interests in recent years and numerous high-quality reviews have been published. However, the filed focused on oral squamous cell carcinoma (OSCC) has not been reviewed and the present article provided supplemental knowledge in this specific cortex.
2)The manuscript contains numerous counts of unclear, ambiguous, and unscientific English writing, as well as grammatic errors and misspellings.

Experimental design

Article content is within the Aims and Scope of the journal. Sources were adequately cited.

Validity of the findings

Conclusion identify unresolved questions and future directions.

Additional comments

Introduction
The first two paragraphs briefly introduced the classification, biosynthesis, sorting, etc. of exosomes, whereas which were repeatedly reviewed by previous manuscripts. Can be shortened.

Reviewer 3 ·

Basic reporting

- The manuscript required an in-depth English editing, as it is often hard to under the meaning of sentences
- References are fine
- The structure of the manuscript needs comprehensive revisions. Each paragraph has similar structure, which consists in a list of examples from literature for the majority of the text and there is not logic transition between different topics (they seems disconnected, stand-alone sections).
For example, in the paragraph 2: lines 122-145, list of proteins in exosomes, lines 145-167 list of effects of exosome proteins on immune cell functions, lines 168-184 list of effects of exosome proteins on metastasis, lines 185-189 effect of exosome proteins on EMT.
As review manuscript on OSCC, a more in-depth critical review and discussion on the role of exosomes in OSCC should be provided. Critical discussion and in depth description of the biological and cellular mechanisms regulated by exosome cargo are required. A particular focus should be directed to the clinical relevance of exosome cargo in OSCC (specific paragraph).

Experimental design

'no comment'

Validity of the findings

'no comment'

Additional comments

The manuscript requires a thorough revision.
The paragraphs need to be reorganized. Their structure requires an elaborated description of the topics with critical comments that are relevant to OSCC. The review manuscript has to deliver important and relevant information on the topic to the reader.

---

## Round 0.2 · accepted · Accept

Thank you for attending to the issues raised under review. The main issue was one of narrative flow, and I think you have made a reasonable job of improving this to a standard such that the story is clear and unambiguous. I am therefore happy to recommend the paper for acceptance now.